# A Tunable Graphene 0–90° Polarization Rotator Achieved by Sine Equation Voltage Adjustment

**DOI:** 10.3390/nano9060849

**Published:** 2019-06-03

**Authors:** Jinsong Dai, Zhongchao Wei, Lin Zhao, Qiyuan Lin, Yuyao Lou

**Affiliations:** Guangdong Provincial Key Laboratory of Nanophotonic Functional Materials and Devices, School of Information and Optoelectronic Science and Engineering, South China Normal University, Guangzhou 510006, China; jsdai@m.scnu.edu.cn (J.D.); 2017021792@m.scnu.edu.cn (L.Z.); 20163280007@m.scnu.edu.cn (Q.L.); 20160121325@m.scnu.edu.cn (Y.L.)

**Keywords:** tunable graphene, polarization rotator, radially polarization converter

## Abstract

Polarization Manipulation has been widely used and plays a key role in wave propagation and information processing. Here, we introduce a polarization rotator in the terahertz range with a polarization conversion ratio up to 99.98% at 4.51 terahertz. It has a single graphene layer on top of the structure patterned by 45° tilted space elliptical rings. By changing the Fermi level from 0.3 ev to 0.7 ev of the graphene, we can turn the reflective light polarization direction between 0° to 90° with nearly unique magnitude. Surface currents theories and graphene characteristics clarify the relationship between polarization angle and Fermi level to be a sine equation adjusted voltage. We firstly put forward an equation to thetunable graphene changing the reflective light polarization angle. It can be widely used in measurement, optic communication, and biology. Besides, with nearly the unique reflective light in different directions, the rotator is designed into a novel radially polarization converter. The latter can be switched from radially polarized light to linearly polarized light, and vice versa, in the terahertz region.

## 1. Introduction

Recently, various periodic metasurface applications achieved absorption [1,2], filter [3], sensing [4] and polarization conversion [5]. In these applications, polarization conversion holds the freedom to wave propagation and information processing. Manipulating wave polarization has been widely used in light generation, sensing, analytical chemistry, imaging, and so on. Based on Faraday effects, polarization conversion of birefringent crystals [6] is too bulky to integrate and miniaturize. By using metallic metasurface structures, some polarization conversions [7,8] overcome the problem, but the material has an invariable parameter that is unable to turn the light freely. Fortunately, graphene, a manipulated material with a two-dimensional plane of carbon atoms has been theoretically and experimentally proved to support surface plasmon polaritons (SPPS) at terahertz (THz) waves, and tuned via biasing the graphene layer to change $\sigma_{graphene}$ [9,10] dynamically. Due to the advantages of the graphene, various manipulating devices included polarization converters. Broadband, ultra-broadband, wide-angel polarization converters [11,12,13,14] have been put forward to achieve a high polarization conversion ratio (PCR). However, their converters are focused on *x-* to *y*- direction polarization conversion and expressed about the little reflective light, which is between the *x-* and *y*- polarization directions. In optical communication, the polar modulation needs various polarization directions. Chen et al. proposed that they can change the polarization angle from 0° to 90° [15]. Magnitude in reflective light is important to integrated optical circuits, which is carefully considered in our work while they did not. Although Fan et al. proposed a polarization rotator [16] which may change polarization direction by rotating the structure, electrical adjustment instead of structure rotation is produced in this work for polarization conversion in terahertz region. Zhang et al.’s polarization rotator can change the reflective polarization angle from 20° to 70° with steady E magnitude [17]. But, among all the devices mentioned above, we first give an exactly equation to describe the relationship between graphene Fermi level and reflective light polarization angle (range of 5.94° to 89.32°).

In the past, radially polarized vector lights were widely used in optical tweezers [18], optical communication [19], super-resolution microscopy [20] high-density optical data storage [21], and so on. After liquid crystal was widely used, liquid-crystal based radially polarized vector light converters were introduced [22]. Because of its bulky volume and the development of nanomaterial, plasmonic metasurface consisting of orthogonal nanoslit pairs to manipulate polarization, [23] was proposed. Indeed, radially polarized vector light is often used in visible light. However, terahertz radiation is not ionizing radiation, and its low photon energies generally do not damage DNA and living tissues [24]. Terahertz radiation can also detect electro-optical (EO) crystal [25] and differences in water content [26]. Radially polarized vector light in THz can help a small focused light to penetrate deeply into the image or detect.

In this letter, we introduce a polarization rotator based on patterned graphene. The unit double elliptical rings on the graphene guide the surface currents. The special surface currents make it not only converts the light polarization direction from 5.94° to 89.32° but also maintains a nearly unique reflective magnitude light in the tunable polarization angle range. A specific equation is firstly produced to calculate the Fermi level with the reflective polarization angle to realize the electrical adjustment due to the $\sigma_{graphene}$. Apart from that, a radially polarized vector light converter is produced. The converter can be switched from radially polarized light to linearly polarized light easily by changing the Fermi level of graphene.

## 2. The Proposed Structure Design and Simulation Results

We choose sandwich structure to design the polarization rotator, with graphene on the top layer, a dielectric layer in the middle, and gold in the bottom of the structure. To realize the function of polarization rotator, two 45° tilted elliptical rings is patterned on the graphene layer periodically. The fabrication errors: ZnSe layer thickness ±1 μm, gold layer thickness ≥1 μm, and graphene ellipses’ long axis and short axises ±0.5 μm. Figure 1b depicts a unit cell of the structure schematically. We assume that the temperature is 300 K. The structure is simulated in Finite-Difference Time-Domain (FDTD). In the simulation, periodic boundary conditions are set in the *x-* and *y-* direction. A plane wave in THz region is vertically incident downward to the structure in the *x*-polarization. Reflective light polarization angle (*θ*) is the acute angle from the reflective polarization direction to incident light polarization direction.

Polarization conversion ratio (PCR) is often used to introduce and manifest the polarization rotator’s performance in THz wavelength [11].
(1)$$PCR=\frac{r_{yx}^{2}}{r_{xx}^{2}+r_{yx}^{2}}$$

In the PCR definition above, $r_{yx}=R_{yr}/R_{xi}$, $r_{xx}=R_{xr}/R_{xi}$ where $R_{yr}$ is the magnitude of reflected light E field in the *y*-direction, $R_{xr}$ is the magnitude of reflected light E field in the *y*-direction, and $R_{xi}$ is the magnitude of incident light E field in the *x*-direction.

The PCR of a polarization converter is above 90% in 4.3–5.2 THz, reaching its peak of 99.98% at 4.51 THz, where the reflective light magnitude is 0.83. And the loss of the light (E) is about 1 dB. Figure 2a shows that polarization converter can turn the light direction from x to y. As demonstrated in Figure 2b, when the graphene Fermi level changes, the PCR curve has an obvious shift. Thus, it has a frequency range where PCR may from nearly 0 to 1 as we adjust the Fermi level.

Next, at 4.51 THz, PCR peaks at 0.3 ev of graphene Fermi level, while reaching nearly 0 at 0.7 ev. Hence, various Fermi levels of the reflective lights are simulated at 4.51 THz to get different PCR values.
(2)$$\theta =\arctan\left(\frac{r_{yx}}{r_{xx}}\right)$$

We turn PCR from 0.01 to 0.99. As a result, we can rotate the reflective polarized linear light angle from 5.94° to 89.32° with the Fermi level ranging from 0.3 ev to 0.7 ev successively, and PCR relates to the reflective light polarization angle. Under this condition, other tunable polarization converters [11] can also turn the PCR from 0 to 1 approximately. This work make further improvement. Firstly, a wider range of Fermi level is provided to change the polarization angle. Secondly, the magnitude of reflective E field is nearly unique about 0.84 (from 0.81 to 0.89) in the tunable ranges showed in Figure 3b. Thirdly, an accurate equation about the relationship between polarization angle and Fermi level is given in this paper which is applicable to integrated optical circuit electric adjustment.

## 3. Discussion on Physical Mechanisms and Theoretical Analyses

Surface currents distributions are used to understand the mechanism in converting the light with nearly unique magnitude.

As displayed in Figure 4a, E fields on the graphene layer go in three different ways due to the $\sigma_{graphene}$. In the graphene gaps, it happens electromagnetic resonances at 4.51 THz. The E direction in the gaps goes differently in the 0.3 ev, 0.51 ev, 0.7 ev. For the inner gaps, E direction goes up-right in the 0.3 ev while in the 0.51 ev and the 0.7 ev, the E direction in the outer gap gose down-right direction which offsets the E magnitude in the inner gaps. Zhu et al.’s sinusoidally-slotted graphene-based cross-polarization converter is regarded as a single ring that has no offset effects. So it can’t rotate the polarization angle, except to 90° [11]. Due to the Maxwell’s equations, the offset effects is well shown in the H (top) fields on the graphene layer which cause the different surface currents intensities. The H magnitude on the bottom fields takes apart in surface currents formation. It is clear that the H currents are weaker when the Fermi level rises. The different electromagnetic resonances cause different magnitudes of the surface currents.

Figure 4b shows the vector synthesis of incident light and the surface currents. The surface currents magnitude determines the reflective light direction, changing from *x*-direction to *y*-direction. When the Fermi level increases, the surface currents (Em) become weaker, so that the reflective E vector turns to another direction. However, not all of the currents on the surface go in this direction, so that reflective light magnitude in the *y*-direction is hard to reach 100%. The surface currents changes can not only switch the light, but also maintain the same electromagnetic resonances model in the graphene gaps, so that the magnitude of reflective light is nearly unique in the range of 0.82–0.89 in Figure 3. It is conformed to $\sqrt{2}E_{90}{=E}_{45}$, in ideal according to the geometric relationship, as the dissipation in 0° and 90° conditions are more than in 45°the lowest is 0.82 in 45° reflective polarization angle. If we change the gaps’ width, the electromagnetic resonances will be stronger or weaker. For example, in Figure 4c, we simulate narrow inner gaps and find that the minimum polarization angle is 8.31°. It is bigger than our proposed structure’s reflective polarization angle (5.94°). Because the stronger electromagnetic resonances make the inner gap E energy stronger, too. The total E energy became stronger than the former. From the vector synthesis of surface currents (Em) and E incidents, it is hard for the latter to have a small reflective polarization angle.

At the same frequency, there is only one current distribution dependent on $\sigma_{graphene}$ ($\sigma_{graphene}=\sigma_{\mathrm{intra}}+\sigma_{iiter}$) for specific reflective polarization angles. The conductivity of the graphene surface is expressed by Kubo formula [27].
(3)$$\sigma_{graphene}\approx \sigma_{inter}\approx \frac{-je^{2}}{4\pi \hbar }ln\left(\frac{2\left|\mu_{c}\right|-\left(\omega -j\tau^{-1}\right)\hbar }{2\left|\mu_{c}\right|+\left(\omega -j\tau^{-1}\right)\hbar }\right)$$
where ω is angular frequency, $\mu_{c}$ is Fermi level, τ is electron scattering time of graphene, and $\tau^{-1}=\frac{ev_{f}^{2}}{\mu_{m}\mu_{c}}$ (4), $v_{f}^{}$ is Fermi velocity, $\mu_{m}$ is Carrier mobility, e is electron charge, and ℏ is the reduced Planck constant. In the THz wavelength range, we have $\hbar \omega \ll 2\left|\mu_{c}\right|$ and $\sigma_{intra}\to 0$. Thus, Equation (3) can be used under this condition. $\sigma_{graphene}$ is considered into a constant with which (3) can be translated into (4), to show the relationship between $\mu_{c}$ and ω.
(4)$$\omega =\alpha \mu_{c}+\beta$$
where α and β are constants.

The frequency ω is almost linear in $\mu_{c}$, as shown in Figure 5a. The simulation results in Figure 5b are based on Equation (3). This special approximate treatment is used to infer that different polarization angles have the same linear formation at different frequencies.

Figure 6a,b demonstrate two conditions at different frequencies. Contrasting the two pictures, plots move to 0.5 ev due to linearity between $\mu_{c}$ and ω. Frequency is changed linearly, as proved in Figure 5 which influences the relationship between polarization angle and Fermi level.

Apart from that, we change the form of (3), which is quoted from [28]:(5)$$\sigma_{graphene}\approx \sigma_{inter}\approx \frac{je^{2}}{4\hbar }\left[\psi \left(\hbar \omega -2\mu_{c}\right)+\frac{1}{\pi }\ln\left|\frac{2\mu_{c}-\hbar \omega }{2\mu_{c}+\hbar \omega }\right|\right]$$
where ψ(x) is the Heaviside step equation. Because of $\hbar \omega \ll 2\left|\mu_{c}\right|$, we simplify (5) and give the relationship between $\sigma_{graphene}$ and $\mu_{c}$ in the constant frequency. We assume $\sigma_{graphene}=\zeta e^{j\theta }$, where ζ is a constant due to the relationship in surface currents on graphene layer:(6)$$\theta =\zeta sin\left(\frac{e^{2}}{4\hbar }\left(\psi \left(\hbar \omega -2\mu_{c}\right)+\frac{1}{\pi }\right)\right)$$

Equation (6) shows the relationship between θ and $\mu_{c}$. The E field changes in the graphene gaps in Figure 4a. It is clear that θ changes by the sine equation, which influences the relationship between reflective polarization angle and Fermi level. Thus, we use the sine equation to speculate the changes in the reflective polarization angles by different Fermi level.
(7)$$\theta =A+Bsin\left(\frac{2\pi }{T}\left(\mu_{c}-\Phi \right)\right)$$
where A and B are calculated by maximum and minimum of θ, Φ changes by wavelength to pan the sine equation caused by the linearity between $\mu_{c}$ and ω, $\frac{T}{2}$ is the wavelength range of $\mu_{c}$ from $PCR_{min}$ to $PCR_{max}$. As shown in Figure 6a, the curve fits well at 4.51 THz. Based on Equation (7), we can control the reflective polarization angle from 0° to 90°.

After getting Equation (7) at 4.51 THz, the curve is panned by cutting 0.5 ev of Φ to fit another mode of turning at 4.65 THz. Because of the limitation of the graphene pattern due to surface currents distribution, the Figure 6b plots do not fit well in the peak of the sine equation (see Figure 2b), where the curve peaks turn down with the increasing frequency. Although imperfection occurs at the peak of the curve, other plots fit well in range of 0.3 ev to 0.7 ev.

It is easy to use the approximate equation to turn the reflective polarization angle to the desired polarization angle. If we get the 4.51 THz change equation, all of the working curves can be inferred by panning the curve.

## 4. Radially Polarized Vector Light Devices

As mentioned above, we take advantages of the polarization converter to design a device to generate radially polarized light. In particular, the device may switch from radially polarized light to linearly polarized light.

The radially polarized converter is set to put polarized rotators in 8 different directions with a THz absorber in the bottom of the structure. Firstly, because of the limitation of a 90-degree range, the gold layer 1/4 λ (about 15 μm) higher than others to make 1/2 λ optical path difference to make the 180° phase different, and the mirror symmetry pattern is created to obtain light in the other directions. Then, the Fermi level of graphene is set to fit (showed on the Figure 7a) in every direction to turn the light to the desired polarization angle by a Multichannel DC voltage (showed on the Figure 7d). Secondly, a small incident angle is set to get reflective light. The incident light angle tolerance of the rotator is shown in Figure 8 with the rotator placed in 8 different directions.

We simulated 3 × 3 units groups on two directions in red circle at 4.51 THz with *x*-direction 15° incident angle plane wave. The up-direction rotator is blocked up to 16.5 μm, so the optical path difference is 1/2 λ which make the up-direction reflective light’s direction opposite to the down-direction rotator reflective light’s. And the 45° tilted-direction rotator has mirror symmetry pattern that causes the reflective direction turn in a symmetry direction. It not only proves that mirror symmetry patterns get another directions, but also that at 1/4 λ higher gold layer can make a 1/2 λ optical path difference.

Apart from that, every Fermi level of the graphene layer is tunable. Hence, if all the graphene layers are turned into 0.7 ev, there are two 180° phase difference linearly polarized lights in the same polarization direction. By changing the Fermi level of the graphene, a new type of switchable radially polarized light converter is proposed.

## 5. Discussion

### 5.1. Fabrication Errors of Dielectric Layer Thickness

The dielectric layer thickness can impact the maximum reflective polarization angle. We simulate the dielectric layer thicknesses of 16 to 19 μm. Because the $r_{xx}$ is down to nearly 0 at a thickness of 17.5 μm in Figure 9a. Apart from $r_{yx}$, PCR curve shift is similar when changing the thickness, and the blue curve, which represents thickness of 17.5 μm, is in the middle of them in Figure 9b,c, so the thickness of 17.5 μm may be the best choice for us. In addition, the fabrication error of the dielectric layer thickness is within ±1 μm from Figure 9.

### 5.2. Integration Schemes

We design two integration schemes for future applying in the photonic integrated circuits. The light from source is adjusted by PC to an *x*-direction linear polarized light. Then, WDM chooses the correct frequency of light. In Figure 10a, after reflected light sent out from the polarization rotator, another PC worked for insulating the incident light and checked the reflective light polarized angle. The coupler and loop-waveguide form a sagnac loop. When the CR is 0.5, the output light is the strongest. In the Figure 10b, with a small incident angle, the incident light and reflected light can be divided.

## 6. Conclusions

To conclude, we have designed a polarization rotator for terahertz region with a high PCR peak. It can turn the reflective polarization angle from 5.94° to 89.32° with a nearly unique magnitude and change Fermi level of the single graphene layer on the structure based on a sine equation which is firstly put forward to the tunable graphene. Additionally, we use surface currents and Kubo formula to explain the specialties of the polarization rotator. Besides, we design a radially polarized light converter which can be switched from radially polarized light to linearly polarized light by changing the graphene Fermi level. This novel and flexible approach is a new way to generate radially polarized light for terahertz region. We believe that the polarization rotator is more accurate and practical in future application and integration.

## Figures and Tables

**Figure 1 nanomaterials-09-00849-f001:**
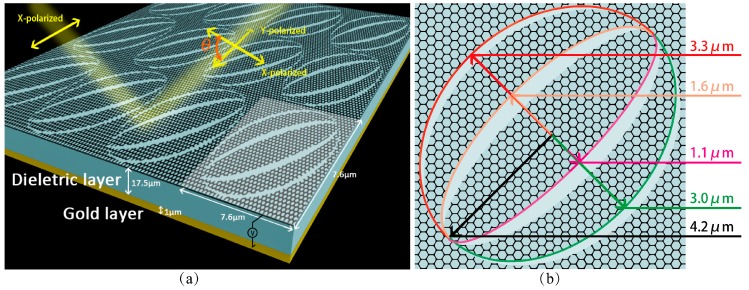
(**a**) Structure of the polarization rotator with three layers, a single and periodic patterned graphene layer on the top of the structure, a 17.5 μm thick dielectric layer of ZnSe (steady index about 2.3 in the mid and far infrared range with Stable chemistry), 1 μm thick gold layer and a DC voltage is added to control devices (graphene Fermi level), which connected with the graphene layer and gold layer. (**b**) A subunit of the rotator, which is a single layer of graphene. The elliptical rings on graphene nested space and graphene ellipses, in which the value of long axis is 4.2 μm and the value of short axis is 3.3 μm, 3.0 μm, 1.6 μm, 1.1 μm, respectively.

**Figure 2 nanomaterials-09-00849-f002:**
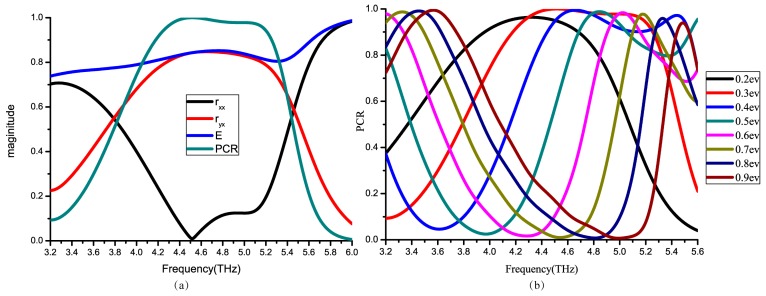
(**a**) The magnitude of $r_{xx}$, $r_{yx}$, E and PCR simulated at $\mu_{c}$ = 0.3 ev. (**b**) PCR changing with different graphene Fermi level.

**Figure 3 nanomaterials-09-00849-f003:**
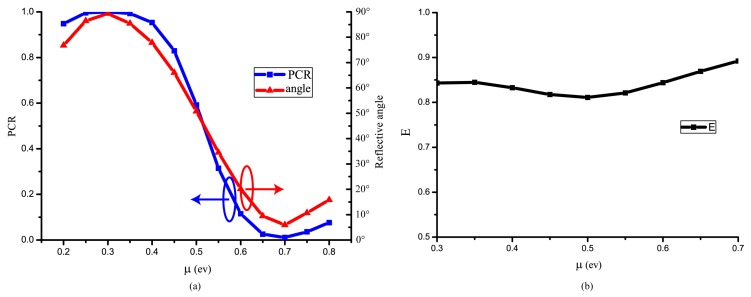
(**a**,**b**) PCR (left), reflective light polarization angle (right), and magnitude of reflective light E field in different graphene Fermi level at 4.51 THz shows in (**a**) and (**b**), respectively. The chemical potential covers all the PCR ranges in (**a**) and the magnitude of reflective light is nearly unique in range of 0.3 to 0.7 ev.

**Figure 4 nanomaterials-09-00849-f004:**
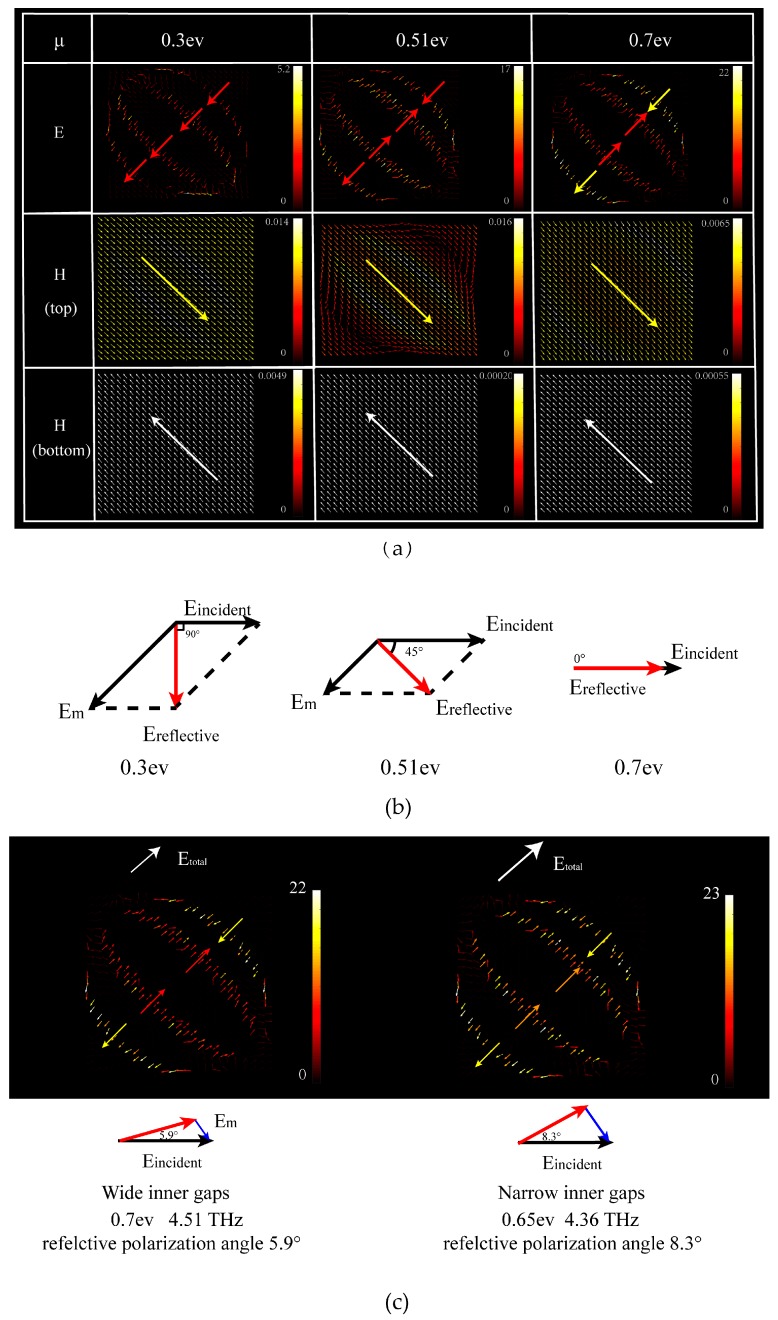
(**a**) Three different reflective polarization angles corresponding to three different Fermi level. E fields on the graphene layer, H field on the graphene layer, and H field on the bottom layer at 4.51 THz, with 0.3 ev, 0.51 ev, 0.7 ev Fermi level, with a reflective light polarization angle of 0°, 45° and 90°, respectively. (**b**) Vector synthesis of incident light and the surface currents at reflective polarization angles of 0°, 45°, and 90°. (**c**) Narrow inner gaps between short axis of 3.3 μm, 3.0 μm, 1.6 μm, 1.2 μm are simulated at 4.51 THz and 0.65 ev graphene Fermi level. Under this condition, the minimum polarization angle is 8.3°.

**Figure 5 nanomaterials-09-00849-f005:**
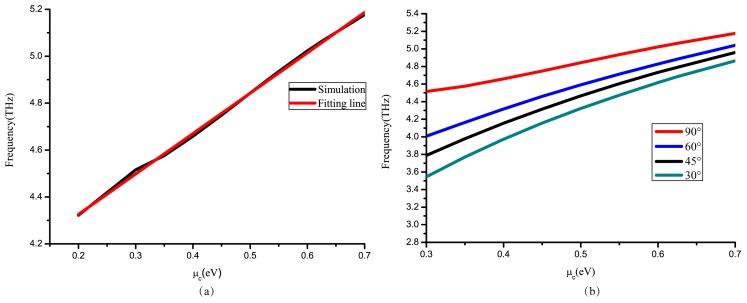
(**a**) When PCR peaks (as the light converting polarization angle reaches 90°), the relationship between frequency and $\mu_{c}$. (**b**) Other polarization angles of reflection light in simulation result.

**Figure 6 nanomaterials-09-00849-f006:**
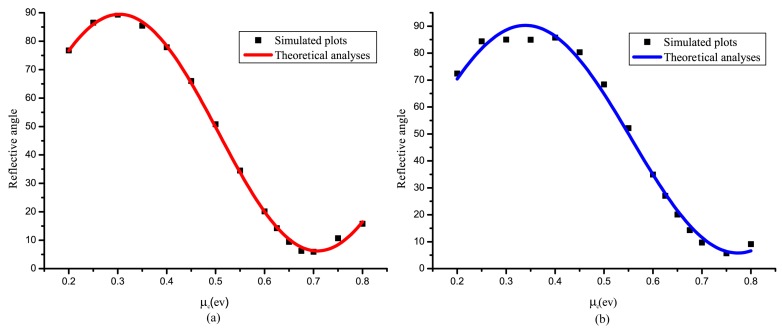
(**a**,**b**) With the changing of $\mu_{c}$, the reflective polarization angle turns from nearly 90° to 5° at 4.51 THz and 4.65 THz, respectively. In theoretical analyses, the sine Equation (7) fit the simulated results.

**Figure 7 nanomaterials-09-00849-f007:**
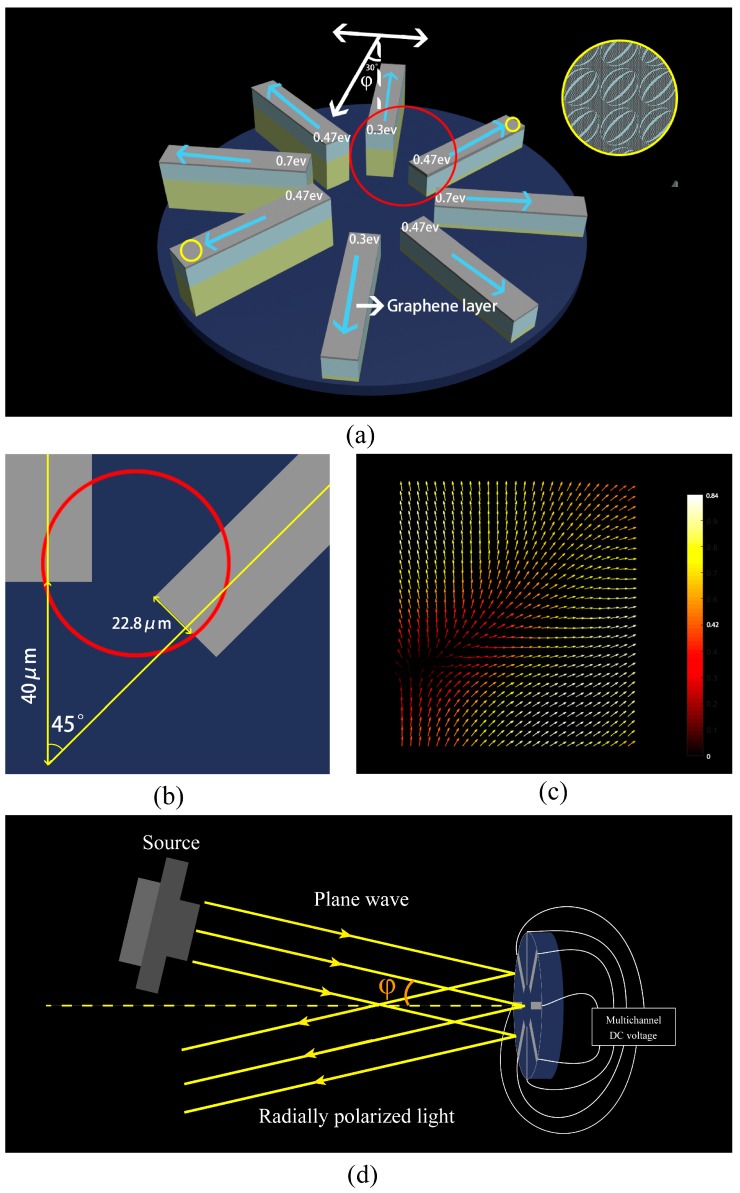
(**a**) Radially polarized light converter with THz absorber (blue) in the bottom of the structure. Eight different directions polarized rotators generate the radially polarized light. The mirror symmetry pattern is in the yellow circle to generate 45° and 235° polarization angle of reflective light. The red circle shows the vector between the two different Fermi level rotators. White arrows mean that incident light angle (φ) is 30° tolerable for polarized light rotating. Blue arrows refer to the vector direction of the reflective light. (**b**) The amplification of the area near the red circle. Red circle area is covered the 22.8 μm rotator width and part of interval space. The width of polarized rotators is 22.8 μm and the distance to the center is 40 μm. (**c**) The red area is simulated to get the reflective E-vector picture of reflective light (**d**). Another view of the radially polarized light converter is to show how it works.

**Figure 8 nanomaterials-09-00849-f008:**
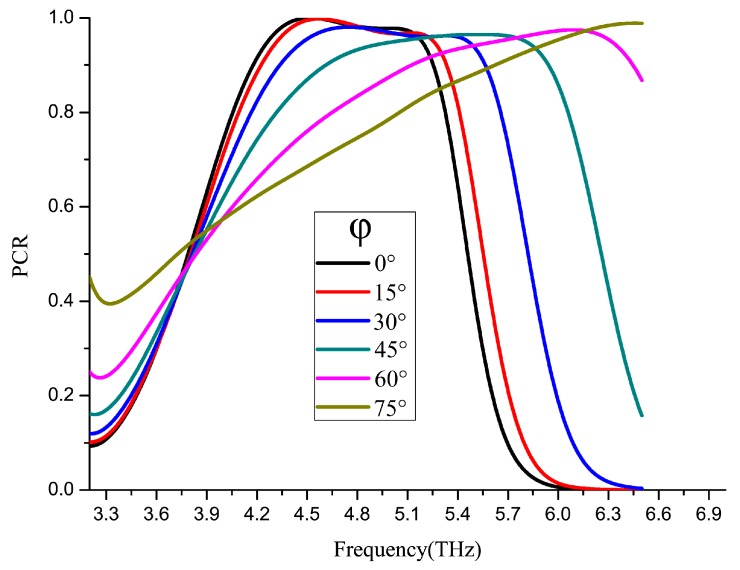
The PCR of the polarization rotator at 0.3ev Fermi level in different incident angles (φ).

**Figure 9 nanomaterials-09-00849-f009:**
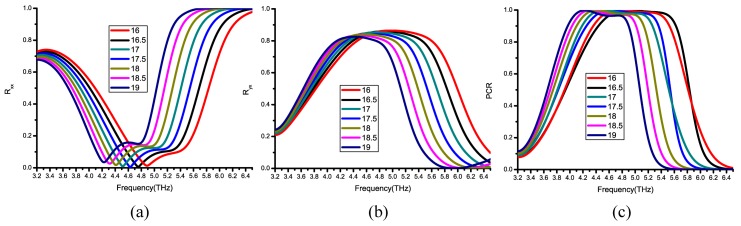
(**a**–**c**) The magnitude of $r_{xx}$, $r_{yx}$ and PCR simulated at $\mu_{c}$ = 0.3 ev in different thicknesses of the dielectric layer.

**Figure 10 nanomaterials-09-00849-f010:**
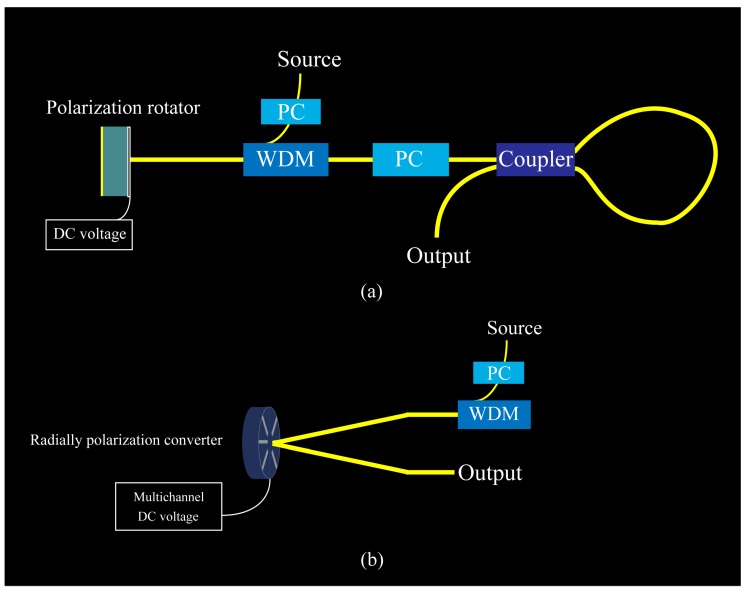
(**a**,**b**) Integration schemes of polarization rotator and radially polarization converter. The yellow line is the Waveguide. WDM is Wavelength Division Multiplexing and PC is the Polarization Controller. The coupling ratio (CR) defined as an the incident light power over the total output power is 0.5.

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
