# Peer review of "A Tunable Graphene 0–90° Polarization Rotator Achieved by Sine Equation Voltage Adjustment"

_nanomaterials, 2019, doi:10.3390/nano9060849_

Author Response

Dear Reviewer:

Thank you for your comments concerning our manuscript entitled “ A Tunable Graphene 0-90° Polarization Rotator Achieved By Sin Equation Voltage Adjustment” (nanomaterials-502035). Those comments are all valuable and very developmental for revising and improving our paper. We have studied all the comments carefully and here below is our description on revision according to your comments in the attachment.

Response: We are appreciated for your positive support to our work. We also make careful revisions about the introduction of our paper, at the same time we have performed more researches and summarized the performance comparisons in the manuscript according to your suggestion. And invited a Doctor in America to correct about 114 grammatical errors and ambiguous phrasings.

Once again, thank you very much for your comments and suggestions!

Sincerely yours,

Jinsong Dai

Reviewer 2 Report

In their paper entitled “A Tunable Graphene 0-90° Polarization Rotator Achieved by Sin Equation Voltage Adjustment,” Dai et al. propose structured graphene for a voltage-controlled polarization-transforming device. Overall, I think the work holds merit as an interesting demonstration of active polarization control and deserves to appear on the pages of Nanomaterials. However, here are the areas I think can be improved:

1. In general, language mistakes made the manuscript difficult to read. The authors need to proof-read the manuscript to eliminate grammatical errors and ambiguous phrasings.

2. Several variables appear in the paper without context or description. E.g, when introducing Eq. 1 for PCR, the authors should include a description of the various variables and indices. They mention only ryx, but should at least also include rxx = Rx,r/Rx,i and explain that indices r and i correspond to the reflected and incident light.

3. There are multiple mix-ups between reflectance, reflection-ratios, and electric-field magnitudes. In Fig 2.a, rxy, ryx, and PCR are plotted but the axis is labeled "E-magnitude" and in Fig. 3b "E" is plotted but axis is titled "magnitude of reflectance". Using the symbol E for electric field instead of R would help eliminate further confusion between E-field and reflectance in line 76.

4. Eq.2, when simplified, reduces to the polarization angle of the reflected light, which should be clearly expressed or stated from the beginning. The presentation of \theta (line 88) is ambiguous; I think the intention was to describe \theta as the difference between the polarization angles of reflected and incident light. A schematic showing \theta, (maybe an addition to Fig.1) would help further clarify the definition.

5. When mentioning any angles, the authors need to emphasize whether they are referencing angle of incidence/reflection, or polarization angles of incident/reflected light. E.g Fig.7 caption "white arrows mean that incident angle is 30° tolerable for polarized light rotating"; or "The  mirror  symmetry pattern is in the yellow circle to generate 45° and 235° light"  required careful reading to understand.

6. The location of the Fig.7c field profile needs clarification.

7. The literature review includes only one reference on graphene-based polarization conversion, despite a lot of recent progress.

8. The discussion on switchability of the radial polarization generator could be improved/expanded. E.g,  the authors need to comment on their simulation beyond "we simulated 3x3 units groups on two directions in red circle";  mention what incident polarization is used; and explain the meaning of "phase difference" in this context.

Author Response

Dear Reviewer:

Thank you for your comments concerning our manuscript entitled “ A Tunable Graphene 0-90° Polarization Rotator Achieved By Sin Equation Voltage Adjustment” (nanomaterials-502035). Those comments are all valuable and very developmental for revising and improving our paper. We have studied all the comments carefully and here below is our description on revision according to your comments in the attachment.

Response: We are appreciated for your positive support to our work. We also make careful revisions about the introduction of our paper, at the same time we have performed more researches and summarized the performance comparisons in the manuscript according to your suggestion. And invited a Doctor in America to correct about 114 grammatical errors and ambiguous phrasings.

Once again, thank you very much for your comments and suggestions!

Sincerely yours,

Jinsong Dai

Round  2

Reviewer 1 Report

I appreciate the modifications of the manuscript done by the authors and for the detailed response letter. I have several comments:
-i think it would be valuable for the manuscript to add to the manuscript the explanation the authors provided in the response letter for the question "how the double elliptical shape is optimized" and a short comparison between double elliptical shape and the shape in ref 11
-add the simulations for ZnSe thicknesses. The authors have the curve as they put it in response letter. This helps also to understand how the device response will change with fabrication errors. 
-i would be better for the comprehension if authors add the figure of 7.) of their response letter as an inset. 

I have also a more serious concern about the integration of the this device with photonic integrated circuits. That is why i raised this question. As the authors only partially replied to this one. It would be of great interest if the authors can provide at small explanation how they want to integrate their device with a photonic integrated circuit. To be more specific, lets imagine a simple case: we have i waveguide, how the authors want to inject the light from the their graphene based polarization rotator to the waveguide? I strongly recommend to address this point if you want the community to consider your device for further integration and not just a stand alone structure. The graphene rotator is more compact than liquid cristal one for exemple but this advantage is important only if it is integrated. If the rotator is bulky but is outside the optical circuit the bulky aspect is much less important.

Author Response

Dear Reviewer:

Thank you for your comments concerning our manuscript entitled “ A Tunable Graphene 0-90° Polarization Rotator Achieved By Sin Equation Voltage Adjustment” (nanomaterials-502035). Those comments are very developmental for revising and improving our paper. We have studied comments carefully and revised the manuscript. And the feedback and revisions are all in the attachment.

Once again, thank you very much for your comments and suggestions!

Sincerely yours,

Jinsong Dai

Round  3

Reviewer 1 Report

I'm globally satisfied how the authors adressed my suggestions. I do not have more comments on the scientific part.
I just advise the authors to read the manuscript again and correct the remaining typos and english. Just a few examples:
line 146-147: "we simulate narrow inner gaps and find the minimum polarization angle is
 8.31° which is bigger than we proposed one"
line 237-238: "And the fabrication errors of the dielectric layer thickness is within  1μm."
line 249 "reflective light"
Missing spaces sometimes  etc.
I let the authors correct this. I do not need to review the new version as the changes will be only related to english or typos

Author Response

Dear Reviewer:
Thank you for your comments concerning our manuscript. We fixed 40 mistakes about gramma and typos. Meanwhile,  we added the missing spaces in the manuscript.

Once again, thank you very much for concerning our manuscript and your comments are all constructive!

Sincerely yours,

Jinsong Dai